# A theory-driven synthesis of symmetric and unsymmetric 1,2-bis(diphenylphosphino) ethane analogues via radical difunctionalization of ethylene

Hideaki Takano [1,2], Hitomi Katsuyama[1,2], Hiroki Hayashi [1,2], Wataru Kanna[3], Yu Harabuchi [1,2,3], Satoshi Maeda [1,2,3,4] ✉ & Tsuyoshi Mita [1,2] ✉

1,2-Bis(diphenylphosphino)ethane (DPPE) and its synthetic analogues are important structural motifs in organic synthesis, particularly as diphosphine ligands with a $C_2$-alkyl-linker chain. Since DPPE is known to bind to many metal centers in a bidentate fashion to stabilize the corresponding metal complex via the chelation effect originating from its entropic advantage over monodentate ligands, it is often used in transition-metal-catalyzed transformations. Symmetric DPPE derivatives ($Ar^1_2P-CH_2-CH_2-PAr^1_2$) are well-known and readily prepared, but electronically and sterically unsymmetric DPPE ($Ar^1_2P-CH_2-CH_2-PAr^2_2$; $Ar^1 \neq Ar^2$) ligands have been less explored, mostly due to the difficulties associated with their preparation. Here we report a synthetic method for both symmetric and unsymmetric DPPEs via radical difunctionalization of ethylene, a fundamental $C_2$ unit, with two phosphine-centered radicals, which is guided by the computational analysis with the artificial force induced reaction (AFIR) method, a quantum chemical calculation-based automated reaction path search tool. The obtained unsymmetric DPPE ligands can coordinate to several transition-metal salts to form the corresponding complexes, one of which exhibits distinctly different characteristics than the corresponding symmetric DPPE–metal complex.

1,2-Bis(diphenylphosphino)ethane (DPPE) and its derivatives are highly important structural motifs in organic synthesis because DPPE acts as a bidentate chelating ligand. Bidentate coordination to a metal center to form five-membered rings provides a distinct entropic advantage relative to monodentate coordination. Therefore, DPPE forms stable transition-metal complexes, especially with late transition metals (group 8–11 elements), which are frequently employed in a wide variety of catalysts[1]. Recently, its dioxide has also become well known for

the synthesis of not only mononuclear metal complexes[2,3], but also coordination polymers assembled with lanthanide metals[4,5]. Several synthetic methods to produce DPPE exist; however, most of these methods are limited to substitution reactions of highly reactive alkali metal diphenylphosphides (e.g., $Ph_2PM$; $M = Li$ or Cs) with 1,2-dihaloethanes ($X-CH_2-CH_2-X$; $X = Cl$, Br, and I) in an $S_N2$ fashion (Fig. 1a, Eq. (1))[6–8]. Moreover, there are a few methods using the alkoxide-base (KOt-Bu)-catalyzed hydrophosphination of vinyl phosphines (Eq. (2))

[1]Institute for Chemical Reaction Design and Discovery (WPI-ICReDD), Hokkaido University, Kita 21, Nishi 10, Kita-ku, Sapporo, Hokkaido 001-0021, Japan. [2]JST, ERATO Maeda Artificial Intelligence in Chemical Reaction Design and Discovery Project, Kita 10, Nishi 8, Kita-ku, Sapporo, Hokkaido 060-0810, Japan. [3]Department of Chemistry, Faculty of Science, Hokkaido University, Kita 10, Nishi 8, Kita-ku, Sapporo, Hokkaido 060-0810, Japan. [4]Research and Services Division of Materials Data and Integrated System (MaDIS), National Institute for Materials Science (NIMS), Tsukuba, Ibaraki 305-0044, Japan. ✉e-mail: smaeda@eis.hokudai.ac.jp; tmita@icredd.hokudai.ac.jp

**a General methods for the synthesis of DPPE**

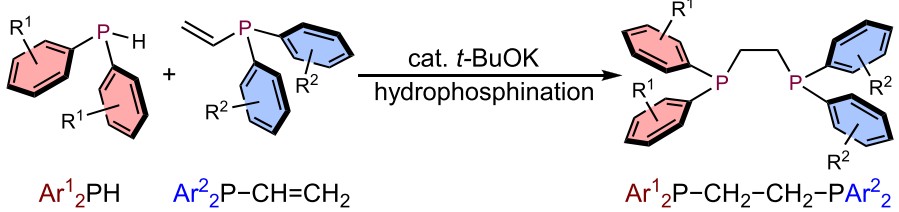

**b The synthesis of unsymmetric DPPE derivatives**

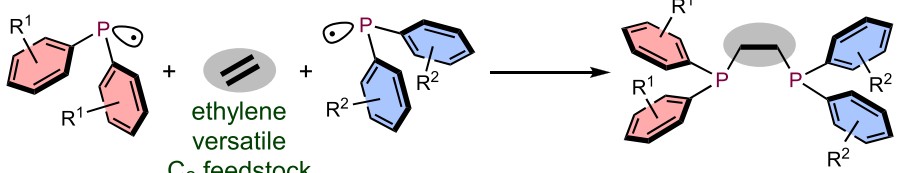

$Ar^1_2PH$        $Ar^2_2P-CH=CH_2$                    $Ar^1_2P-CH_2-CH_2-PAr^2_2$

■ Limited synthetic methods  ■ Multi-step synthesis of starting materials

**c Steric and electronic consideration in (un)symmetric DPPE derivatives**

Symmetric            Unsymmetric

electron-donating    electron-withdrawing

EDG                  EWG

EDG    $R^1$  $R^2$   EWG

steric effects

**d This work: the synthesis of DPPEs from ethylene and two P-centered radicals**

ethylene
versatile
$C_2$ feedstock

■ Readily available starting materials  ■ Facile synthesis of (un)symmetric DPPEs

**Fig. 1 | Synthetic strategies and the design of (un)symmetric DPPEs. a** Synthesis of symmetric and **b** unsymmetric DPPE derivatives. **c** Steric and electronic considerations in (un)symmetric DPPE derivatives. EDG electron-donating group, EWG electron-withdrawing group. **d** The synthesis of DPPEs from ethylene and two phosphine-centered radicals.

with diphenylphosphines[9] or the dihydrophosphination of acetylene gas (Eq. (3))[10]. However, these methods require unstable and highly reactive organometallic phosphorus compounds, which limits their functional-group tolerance and leads to undesired regioselectivity in the hydrophosphination.

Moreover, unsymmetric DPPE derivatives ($Ar^1_2P-CH_2-CH_2-PAr^2_2$; $Ar^1 \neq Ar^2$) are difficult to prepare using general $S_N2$ substitutions, because the selectivity of the substitution is problematic. To synthesize such unsymmetric DPPE derivatives, hydrophosphinations are generally employed (Fig. 1b)[11-16]. However, the preparation of a diarylphosphine ($Ar^1_2PH$) and a diarylvinylphosphine ($Ar^2_2P-CH=CH_2$) with different substituents is required, each of which usually involves multiple synthetic steps. Therefore, it has been difficult to broaden the versatility of the synthesis of unsymmetric DPPE derivatives via hydrophosphinations. Despite the difficulties associated with their

preparation, unsymmetric DPPEs sometimes improve the reactivity, regioselectivity, and even enantioselectivity of transition-metal-catalyzed reactions compared to symmetrical ligands[14,15,17-19]. The advantage of these unsymmetric ligands is the ability to control their electronic and steric properties via judicious choice of the substituents on the phosphorus atoms (Fig. 1c). Phosphorus atoms that bear two electron-donating aryl groups, which often contain bulky alkyl substituents, in general strongly coordinate to metal centers. On the other hand, phosphorus atoms that bear two electron-withdrawing aryl groups usually exhibit weaker coordination. The deliberate introduction of phosphine-substituent combinations that are characterized by push–pull/big–small properties can thus be expected to contribute to reactivity/selectivity patterns. For example, unsymmetric DPPEs change the coordination pattern of Rh/Ir catalysts to improve the linear/branched selectivity in the hydroformylation of 1-hexene[14].

Moreover, unsymmetric bidentate 2,2′-bis(diphenylphosphino)−1,1′-binaphthyl (BINAP) ligands also greatly enhance the reactivity of their transition-metal complexes compared to those carrying symmetric ligands[19]. Therefore, the replacement of symmetric with unsymmetric ligands has become a powerful strategy to improve the reactivity/selectivity pattern of target reactions. The successful applications reported thus far have encouraged us to devise a robust synthetic strategy for symmetric and unsymmetric DPPE ligands from readily available starting materials such as ethylene. If the two carbon atoms of ethylene could react with two phosphine-centered radicals, it would represent a facile and practical method for the synthesis of symmetric/unsymmetric DPPE derivatives (Fig. 1d).

Ethylene is a fundamental two carbon source ($C_2$ unit) that is mostly used as a monomer for the industrial production of polyethylene[20,21]. Ethylene is also used for the synthesis of vinyl chloride, which in turn serves as a monomer for the industrial production of polyvinylchloride, as well as for the synthesis of ethylene oxide and acetaldehyde, both of which are used as starting materials for fine chemicals[22,23]. Although numerous transition-metal-catalyzed transformations such as hydrovinylation, alkene metathesis, and cyclization have been extensively studied on the laboratory scale[24–30], examples for the radical difunctionalization of ethylene[31–36] remain very limited, and only a few examples, including our recent results[36], have been reported to date. This motivated us to explore a different type of radical difunctionalization of ethylene to synthesize DPPE derivatives. Reported examples of the synthesis of 1,2-*bis*(diphosphino)ethane skeletons from ethylene employ diphosphines as a starting material[37–40]. However, these examples require relatively harsh conditions ($T > 200\,°C$ or irradiation with UV light in the presence or absence of a catalytic amount of iodine). Moreover, the substituents on the phosphorus atoms are limited to Me, F, and Cl groups. To the best of our knowledge, reports on the preparation of DPPE derivatives from ethylene in which aryl groups (Ar) are attached at the phosphorus atoms remain unexplored. In this work, we disclose a facile and practical synthetic method for both symmetric and unsymmetric DPPEs from two phosphorous compounds with ethylene by the aid of the artificial force induced reaction (AFIR) method[41], which is a quantum chemical calculation-based automated reaction path search tool. In addition, the obtained unsymmetric DPPE ligands can be used for the synthesis of several transition-metal complexes such as Ni, Pd, Pt and Au complexes, one of which shows an unambiguously different photophysical property compared to the corresponding symmetric DPPE complex.

## Results and discussion
### Quantum chemical calculations for the synthesis of DPPE
To confirm the feasibility of our synthetic approach, we calculated a reaction diagram for the synthesis of DPPE as the target compound via a quantum chemical calculation-aided backward reaction path search using the AFIR method in the global reaction route mapping (GRRM) program[41] combined with the Gaussian 16 program[42]. In this study, artificial forces to cleave the C−P bond were added on both the phosphorus and the carbon atoms (Fig. 2a). These pulling forces induce the formation of two radicals via homolytic cleavage of the C−P bond, and then the backward reaction path search was automatically continued to find new equilibrium structures that have higher energy than the starting DPPE. The obtained equilibrium structures should include candidate starting materials for the synthesis of DPPE. The calculations were carried out at the UωB97X-D/Def2-SV(P) level of theory in dichloromethane (CPCM model), using Grid = FineGrid, charge = 0, and spin = singlet with the option Stable = Opt, which specifies that an MO-stability check is performed at some geometries. After the locally updated planes (LUP) path was estimated, the obtained structures were re-optimized at the UωB97X-D/Def2-SV(P) level with Grid = UltraFine in dichloromethane (CPCM model) using charge = 0 with both spin states (spin = singlet with Stable = Opt, and spin = triplet) (Fig. 2b).

In our retrosynthetic approach, DPPE was successfully separated into ethylene and diphosphine ($Ph_2P−PPh_2$) with reasonable values for the transition state and intermediate energies; the total energy of ethylene and $Ph_2P−PPh_2$ is 17.4 kcal/mol higher than that of the starting DPPE. In this retrosynthetic study using AFIR, the C−P bond of DPPE was cleaved homolytically to provide the phosphinyl radical ($Ph_2P·$) and the alkyl radical (**INT-I-triplet**). Subsequent elimination of the phosphinyl radical led to the generation of ethylene and another phosphinyl radical (**INT-II-triplet**) through **TS-I-singlet** or **TS-I-triplet**, whereby the activation barriers for the singlet state and a triplet state are almost identical. Finally, the diphosphine was obtained via radical–radical coupling between two phosphinyl radicals (**SM-singlet**). In the calculations on the singlet states (**INT-I-singlet** and **INT-II-singlet**), equilibrium structures could not be obtained due to the barrierless radical–radical coupling for the formation of DPPE and **SM-singlet**, respectively. On the other hand, when the triplet state of the diphosphine was calculated starting from the optimized structure in the singlet state, the P−P single bond was automatically cleaved to form two phosphinyl radicals. This observation indicates that, if homolytic radical cleavage of $Ph_2P−PPh_2$ could proceed experimentally, the reaction of diphosphine and ethylene could be expected to proceed smoothly to afford DPPE.

There are several precedents for reactions between diphosphines and alkenes under radical conditions[43–45]. Hirano, Miura, and co-workers have reported alkene difunctionalizations using diphosphines by combining a photoredox catalyst and *N*-bromosuccinimide (NBS)[44]. According to their report, cleavage of the P−P single bond could occur via treatment of the diphosphine with NBS to generate diaryl-bromophosphine ($Ar_2PBr$), followed by reduction induced by the photoredox catalyst to form the phosphinyl radical and a bromide anion. However, the substrate scope is limited to styrene derivatives and aliphatic alkenes are not applicable. Kawaguchi, Ogawa, and co-workers have developed a selective phosphinylphosphination of alkenes with diphosphine monoxides triggered by a radical initiator or photoirradiation with a Xe lamp (>300 nm)[43]. These authors have also reported the thiophosphorylphosphination of hydrofuran to synthesize unsymmetric variants[45]. However, only three unsymmetric examples were reported in this paper, and there are no reports on the use of ethylene as the alkene part. Accordingly, one could feasibly conclude that there is significant demand for the development of more general and diverse methods for the synthesis of symmetric and unsymmetric DPPE derivatives from ethylene under mild conditions without using UV light.

Based on our calculation results shown in Fig. 2b, the energy difference between the ground state and the triplet state of diphosphines was found to be smaller ($\Delta E_{\text{INT-II-triplet/SM-singlet}} = 39.0$ kcal/mol) than the energy required by the representative Ir-based photocatalyst $Ir(ppy)_3$ to reach of the triplet state ($\Delta E = 58.1$ kcal/mol)[46]. These results strongly indicate that the P−P single bond could be cleaved by energy transfer from the photocatalyst without using any additional promoters such as NBS. Once the P−P single bond is cleaved, the thermodynamically more stable DPPE can be expected to be generated following the calculated diagram, in which the energy difference between **INT-I-triplet** and **INT-II-triplet** is very small (1.2 kcal/mol) with a very low activation barrier (9.9 kcal/mol).

### Experimental verification and condition screening
We first examined the reaction using $Ph_2P−PPh_2$, which must be prepared in a glove box due to its high reactivity toward atmospheric air and moisture. The radical reaction was conducted under 10 atm of ethylene gas in the presence of $Ir(ppy)_3$ as a photocatalyst and irradiation from blue LEDs (440 nm) (Fig. 2c, Eq. (1)). Under these conditions, $Ph_2P−PPh_2$ reacted smoothly with ethylene to afford DPPE, which was isolated in 82% yield after protection with $BH_3$·THF. Since $Ph_2P−PPh_2$ is readily prepared from $Ph_2PH$ and $Ph_2PCl$ without a base,

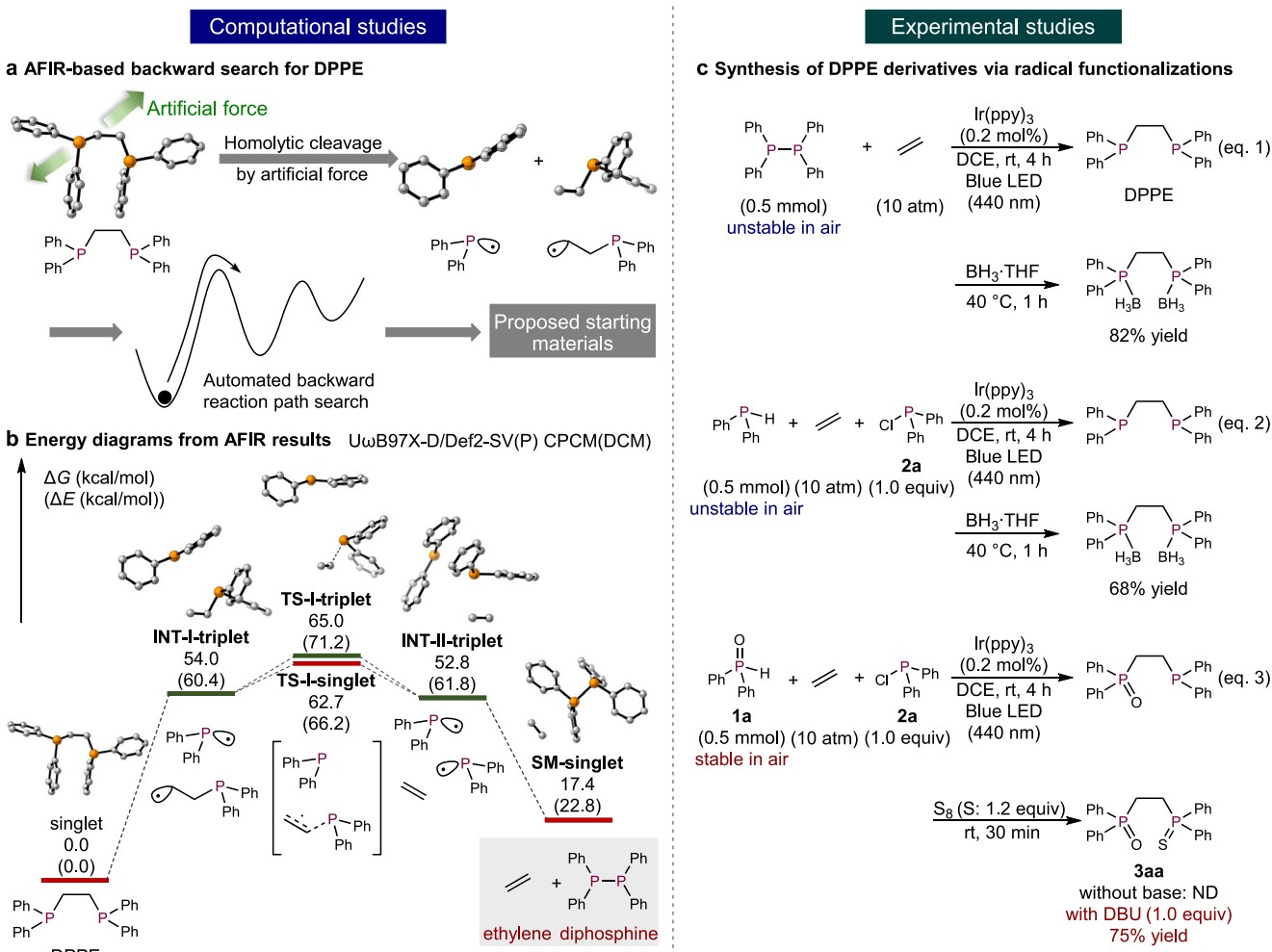

**Fig. 2 | The AFIR-based reaction development for the synthesis of DPPEs.**
**a** Overview of the AFIR-based backward search for DPPE. **b** Retrosynthetic analysis of DPPE using AFIR. Thick red and green lines represent the calculation results for singlet or triplet states, respectively. Hydrogen atoms are omitted for clarity. **c** Synthesis of DPPE derivatives via radical functionalizations. DCM dichloromethane, ppy 2-phenylpyridine, DCE 1,2-dichloroethane.

these two components were then mixed under 10 atm of ethylene, affording DPPE in 68% yield after protection (Eq. (2)). Since Ph₂PH is also unstable in air, the use of a glove box was still required. Thus, the corresponding Ph₂P(O)H (**1a**), which is chemically stable in air, was next employed as a substrate without using a glove box. Although the reaction did not proceed without a base, the expected three-component reaction (3CR) took place efficiently in the presence of 1 equiv of 1,8-diazabicyclo[5.4.0]undec-7-ene (DBU), affording DPPE in 75% yield after oxidation of the remaining phosphorus atom with S₈ (Eq. (3)).

Given the promising experimental results for the synthesis of DPPE derivatives from ethylene via the 3CR, we commenced screening of the conditions using 0.1 mmol of substrates **1a** and **2a** to optimize the reaction conditions (Table 1). We first screened various photocatalysts (1 mol%) under irradiation from blue LEDs. Among the various Ir catalysts employed, both Ir(ppy)₃ and [Ir(ppy)₂(dtbbpy)]PF₆ exhibited high catalytic activity, although [Ir{dF(CF₃)ppy}₂(dtbbpy)]PF₆ did not, probably due to oxidative decomposition of the trivalent phosphorus compounds as a result of its high oxidation potential (entries 1–3)[47]. The use of the Ru-based photocatalyst [Ru(bpy)₃](PF₆)₂ produced the desired product **3aa** in merely 3% yield because several unknown compounds were generated (entry 4). A thermal reaction under complete exclusion of light also promoted the 3CR, whereby **3aa** was formed in 56% yield, indicating that the thermal process contributes to some extent (vide infra), albeit that the reaction is

significantly accelerated by photoirradiation in the presence of a catalyst (entry 5). Among the photocatalysts examined, we chose [Ir(p-py)₂(dtbbpy)]PF₆ as the optimal catalyst for a further screening of the reaction conditions (entry 3). When the pressure of ethylene was reduced to 5 atm or 1 atm (balloon), the reaction still proceeded, albeit in lower yields compared to the reaction under 10 atm (entries 6 and 7). Finally, we succeeded in reducing the catalyst loading to 0.2 mol% on the 0.5 mmol scale, which afforded **3aa** in 82% isolated yield (entry 8). Interestingly, the 3CR proceeded even in the absence of the photocatalyst under irradiation from blue LEDs, even though the yield decreased to 74% yield (entry 9). White LED light, which is a user-friendly light source that covers all wavelengths of the visible-light region, also promotes the 3CR in 65% yield, even in the absence of a photocatalyst (entry 10). Finally, a preparative-scale (0.5 mmol) reaction conducted under irradiation from white LEDs gave a slightly higher yield by prolonging the reaction time to 24 h (entry 11). These experimental results indicate that the use of a photocatalyst is preferable under irradiation from blue LEDs. However, white LEDs also significantly promote the 3CR efficiently, even in the absence of a photocatalyst.

### Scope of symmetric DPPE and application to unsymmetric DPPE
Then, we investigated the substrate scope for the synthesis of symmetric DPPE derivatives under two different sets of reaction conditions: A) in the presence of an Ir photocatalyst under irradiation from

## Table 1 | Optimizing the reaction conditions

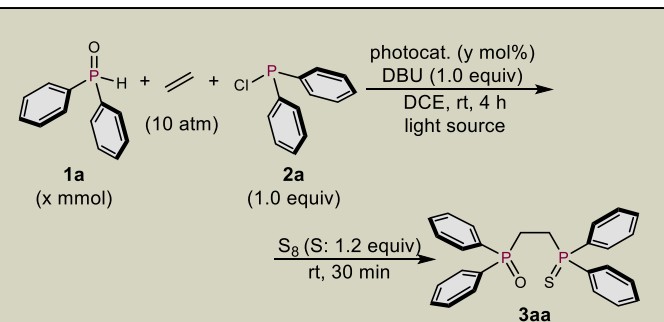

| Entry | Photocatalyst | x | y | light | yield[a] (%) |
|---|---|---|---|---|---|
| 1 | Ir(ppy)₃ | 0.1 | 1 | Blue | 90 |
| 2 | [Ir{dF(CF₃)ppy}₂(dtbbpy)]PF₆ | 0.1 | 1 | Blue | 33 |
| 3 | [Ir(ppy)₂(dtbbpy)]PF₆ | 0.1 | 1 | Blue | 90 |
| 4 | [Ru(bpy)₃](PF₆)₂ | 0.1 | 1 | Blue | 3 |
| 5 | [Ir(ppy)₂(dtbbpy)]PF₆ | 0.1 | 1 | – | 56 |
| 6[b] | [Ir(ppy)₂(dtbbpy)]PF₆ | 0.1 | 1 | Blue | 81 |
| 7[c] | [Ir(ppy)₂(dtbbpy)]PF₆ | 0.1 | 1 | Blue | 78 |
| 8 | [Ir(ppy)₂(dtbbpy)]PF₆ | 0.5 | 0.2 | Blue | 88 (82[d]) |
| 9 | – | | 0.1 | Blue | 74 |
| 10 | – | | 0.1 | White | 65 |
| 11[e] | – | | 0.5 | White | (74[d]) |

Conditions: photocatalyst (y mol%), 1a (x mmol), 2a (x mmol), DBU (1.0 equiv), ethylene (10 atm), DCE (0.33 M), rt, 4 h, and two LEDs

[a]Yields were determined via ¹H NMR analysis using 1,1,2,2-tetrachloroethane as the internal standard

[b]5 atm of ethylene was used

[c]1 atm of ethylene (balloon) was used

[d]Isolated yields

[e]Reaction time: 24 h. ppy: 2-phenylpyridine, dF(CF₃)ppy: 2-(2,4-difluorophenyl)-5-trifluoromethylpyridine, dtbbpy: 4,4'-bis(1,1-dimethylethyl)-2,2'-bipyridine, bpy: 2,2'-bipyridine

blue LEDs (440 nm) for 4 h, and B) in the absence of a photocatalyst under irradiation from white LEDs for 24 h (Fig. 3). Not only electronically neutral **3aa**, but also highly electron-deficient **3bb**, which bears *p*-CF₃-phenyl substituents on the phosphorus atoms, were obtained in good yield under either set of conditions. Products **3cc** and **3dd**, which bear *p*-Cl-phenyl groups or furan rings on the phosphorus atoms, were also obtained in moderate yield. The 3,5-di-*tert*-butyl-4-methoxyphenyl (DTBM) group, which is an interesting moiety due to its steric bulk and high electron-donating properties, promoted the 3CR to afford **3ee**, albeit that the yields were low regardless of the light source employed. Nevertheless, the bidentate phosphine ligands derived from **3ee** (vide infra) can be expected to be very useful, considering the potential of the ligand-substrate dispersion interactions caused by the bulky and hydrophobic alkyl substituents of the DTBM groups[48]. The *p*-OMe-phenyl substituent on the phenyl ring also promoted the 3CR under irradiation from blue LEDs to afford electron-donating DPPE derivative **3ff**.

Having successfully investigated the scope of symmetric DPPE derivatives, we then extensively studied the synthesis of unsymmetric DPPE derivatives using different phosphine oxides (Ar¹₂P(O)H) and chlorophosphines (Ar²₂PCl; Ar¹ ≠ Ar²) under 10 atm of ethylene (Fig. 4). We first examined electron-donating (*p*-NMe₂-C₆H₄)₂P(=O)H (**1g**) and electron-deficient (*p*-CF₃-C₆H₄)₂PCl (**2b**) under conditions A and B. Gratifyingly, unsymmetric DPPE derivative **3gb** was successfully obtained in at least 63% yield under either set of reaction conditions. Examples of the synthesis of DPPE derivatives that bear *p*-NMe₂-phenyl and *p*-CF₃-phenyl substituents as the aryl groups have not yet been reported, which highlights the utility of this 3CR strategy using ethylene. As previous examples of the synthesis of electronically

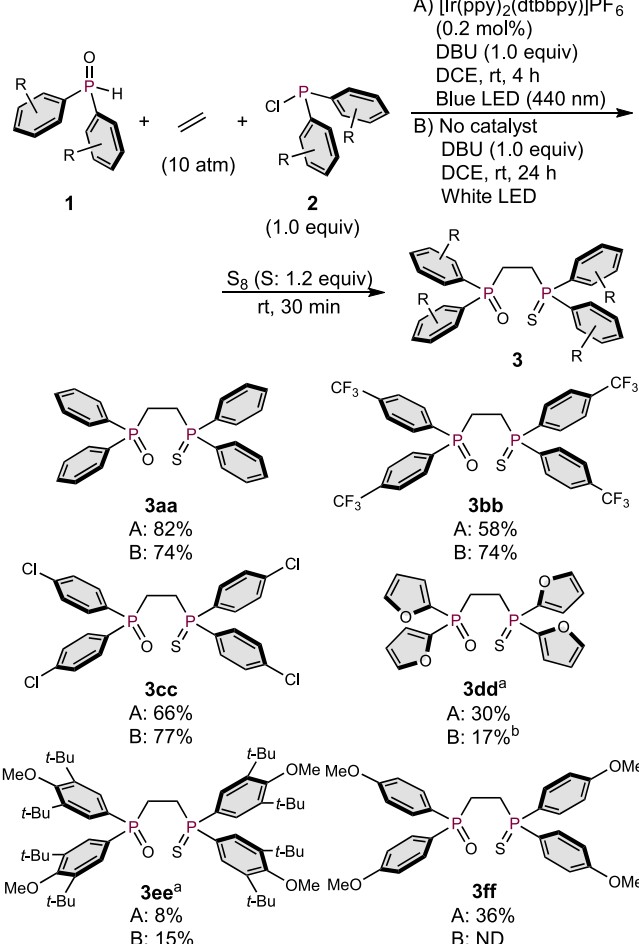

**Fig. 3 | Scope of symmetric DPPE derivatives.** Conditions A: Ir catalyst (0.001 mmol), **1** (0.5 mmol), **2** (0.5 mmol), DBU (0.5 mmol), ethylene (10 atm), DCE (1.5 mL), and two blue LEDs. Conditions B: **1** (0.5 mmol), **2** (0.5 mmol), DBU (0.5 mmol), ethylene (10 atm), DCE (1.5 mL), and two white LEDs. ªReaction scale: 0.25 mmol. ᵇOnly one white LED was employed because two LEDs did not provide the product.

unsymmetric compounds have primarily employed base-mediated hydrophosphinations of vinylphosphines (vide supra)[11–16], the operational simplicity of our method, which simply involves the mixing of three readily accessible starting materials, represents a significant advantage compared to previously reported approaches. Encouraged by this successful first trial of the synthesis of unsymmetric DPPEs, we continued to extend the substrate scope. Electron-withdrawing (*p*-CF₃-C₆H₄)₂PCl (**2b**) was chosen next as the chlorophosphine unit to initially screen the phosphine oxide part. 3CRs with OMe-phenyl groups proceeded in good yield regardless of the position of the substituent (**3fb**, **3hb**, and **3ib**) under either set of reaction conditions. When the 3CR with **1j**, which contains a chiral center on the phosphorus atom, was conducted, racemic **3jb** was obtained in high yield, and both enantiomers were successfully separated by chiral HPLC for further use as chiral phosphine ligands (for details, see Supplementary Figs 2–4). The electron-donating phosphine oxides **1k** and **1e**, which bear *t*-butyl groups on the phenyl rings, also efficiently promoted the 3CR to furnish **3kb** and **3eb**, respectively. Electron-withdrawing *p*-F-phenyl and *p*-Cl-phenyl substituents afforded **3lb** and **3cb**, respectively. Simple phenyl and naphthyl rings promoted the 3CR with high efficiency (**3ab**, **3mb**, and **3nb**). Next, the electron-donating (*p*-NMe₂-C₆H₄)₂P(=O)H (**1g**) was fixed as the phosphine oxide unit for further screening of various chlorophosphines. Substituents with electron-neutral, -withdrawing (*p*-F, *p*-Cl, and 3,5-diCF₃), or -donating (*p*-Me and 3,5-diMe)

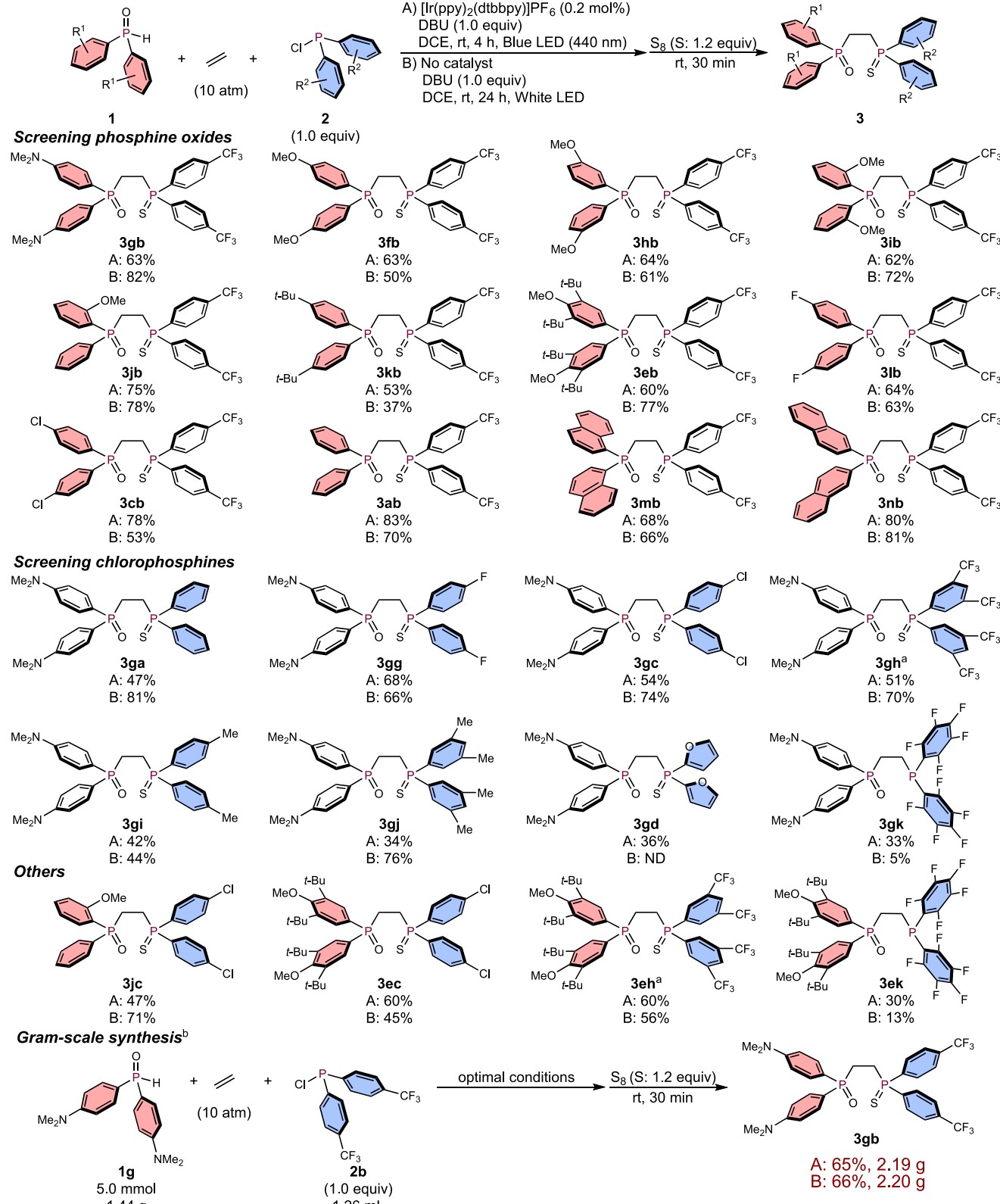

**Fig. 4 | Scope of unsymmetric DPPE derivatives.** As with Fig. 3. [a]Sulfur (2.4 equiv) was added, and the reaction mixture was stirred for 24 h. [b]Four white LEDs was used for condition B.

groups on the phenyl rings are suitable for the 3CR to afford the desired products (**3ga**, **3gg**, **3gc**, **3gh**, **3gi**, and **3gj**) in moderate to high yield under either set of reaction conditions. Unsymmetric DPPEs with other aromatic substituents, such as furan, could be synthesized under irradiation from blue LEDs, while white LEDs did not afford the

targeted product **3gd**. The 3CR with $(C_6F_5)_2$PCl (**2k**) afforded the corresponding product, albeit that it was not oxidized by sulfur ($S_8$) due to the electron-deficient character of the phosphorus atom; thus, **3gk** was directly purified and isolated without prior treatment with sulfur. To further expand the generality of the 3CR, further

**Table 2 | Ethylene Insertion into Diphosphine 4aa and 4gb**

**4aa** (R$^1$ = R$^2$ = H)
**4gb** (R$^1$ = NMe$_2$, R$^2$ = CF$_3$)

**3aa** (R$^1$ = R$^2$ = H)
**3gb** (R$^1$ = NMe$_2$, R$^2$ = CF$_3$)

**4aa'** (R$^1$ = R$^2$ = H)
**4gb'** (R$^1$ = NMe$_2$, R$^2$ = CF$_3$)

| Entry | Compound | Photocatalyst | Light | Temp (°C) | C$_2$H$_4$ (atm) | 3$^a$ (%) | 4$^{ra}$ (%) |
|---|---|---|---|---|---|---|---|
| 1 | **4aa** | [Ir(ppy)$_2$(dtbbpy)]PF$_6$ | Blue | rt | 10 | 85 | – |
| 2 | **4aa** | – | Blue | rt | 10 | 74 | 12 |
| 3 | **4aa** | – | White | rt | 10 | 71 | 15 |
| 4 | **4aa** | – | – | rt | 10 | 50 | 24 |
| 5 | **4aa** | – | – | 50 | 10 | 40 | 26 |
| 6 | **4aa** | – | – | rt | 1 | 49 | 27 |
| 7 | **4gb** | [Ir(ppy)$_2$(dtbbpy)]PF$_6$ | Blue | rt | 10 | 64 | – |
| 8 | **4gb** | – | White | rt | 10 | 80 | – |
| 9$^b$ | **4gb** | – | – | rt | 10 | 35 | 15 |

$^a$Yields were determined via $^1$H NMR analysis using 1,1,2,2-tetrachloroethane as the internal standard
$^b$Diphosphine **4gb** was recovered in 34% yield together with its oxidized analog **4gb'** in 15% yield (total 49% recovery)

combinations of phosphine oxides and chlorophosphines were investigated. The reaction of phosphine oxide **1j**, which bears a chiral center, and (*p*-Cl-C$_6$H$_4$)$_2$PCl (**2c**) furnished **3jc** in good yield. Unsymmetric DPPE derivatives **3ec**, **3eh**, and **3ek**, which contain bulky and electron-donating DTBM groups, were also successfully obtained irrespective of the character of the chlorophosphine, and **3ek** was isolated without the need for sulfur treatment. Moreover, the 3CR could be successfully applied to the gram-scale synthesis of **3gb**. When the 3CR was conducted on the 5-mmol scale, i.e., using 1.44 g of **1g** and 1.26 mL of **2b**, 2.19 g (65% yield) and 2.20 g (66% yield) of **3gb** were isolated under conditions A and B, respectively. As for other alkene counterparts, 1-hexene and styrene did not provide the desired product both under conditions A and B (3 equiv of alkene was used under 1 atm of nitrogen atmosphere), which have significantly highlighted the use of ethylene for this 3CR. We also examined the 3CR of dialkylphosphine oxides (*n*-butyl, *tert*-butyl, and cyclohexyl), chlorodiarylphosphines, and ethylene; however, these reaction systems were complicated and the target compounds were not obtained.

**Mechanistic investigation**

Having examined the substrate scope of the 3CR, we then devoted significant effort to revealing the underlying reaction mechanism based on experimental and computational considerations. To detect the key intermediate of the 3CR, Ph$_2$P(O)H (**1a**) and Ph$_2$PCl (**2a**) were mixed in the presence of 1 equiv of DBU. Instant monitoring by $^{31}$P NMR spectroscopy indicated that Ph$_2$P(=O)−PPh$_2$ (**4aa**) was generated in 93% yield, and the $^{31}$P NMR chemical shift was consistent with literature values (See Supplementary Figs 5, 6)[49]. To examine an unsymmetric variant, a mixture of (*p*-NMe$_2$-C$_6$H$_4$)$_2$P(=O)H (**1g**) and (*p*-CF$_3$-C$_6$H$_4$)$_2$PCl (**2b**) in the presence of 1 equiv of DBU was monitored by $^{31}$P NMR spectroscopy, which indicated the formation of diphosphine **4gb** in 95% yield. Based on these experiments, we conclude that diphosphines **4** are initially generated in situ and should thus be considered as the intermediates of the 3CR.

Given the experimental results regarding the intermediates in the 3CR, we were thus motivated to synthesize intermediates **4aa**[43] and **4gb** to investigate their reactivity not only under LED irradiation (blue or white, with/without photocatalyst), but also under thermal

conditions with complete exclusion of ambient light for 4 h (Table 2). Under irradiation from blue LEDs in the presence of the optimized Ir catalyst ([Ir(ppy)$_2$(dtbbpy)]PF$_6$) under 10 atm of ethylene, the incorporation of ethylene into **4aa** was promoted very efficiently (85% yield), which is in accordance with the result from the 3CR (entry 1 vs entry 3 in Table 1). Furthermore, blue LEDs and white LEDs in the absence of a photocatalyst also promoted the reaction in slightly decreased yield (74 and 71% yield, respectively), which is similar to the outcomes from the 3CR (entries 2 and 3 vs entries 9 and 10 in Table 1). However, very interestingly, without any catalyst and additive and under complete shielding from light, the ethylene insertion proceeded at room temperature under 10 atm of ethylene pressure (50% yield), albeit that the efficiency is low compared to that under photoirradiation, and the starting material was recovered in 24% yield in the oxidized form (**4aa'**; entry 4). The 3CR without light (entry 5 in Table 1) contains a stoichiometric amount of DBU·HCl, which has the potential to promote non-light-driven reactions. Nevertheless, an experiment using isolated **4aa** completely discarded such an effect of DBU·HCl and allowed the conclusion that a thermal reaction should proceed irrespective of the presence or absence of any additive and catalyst. Increasing the temperature to 50 °C slightly decreased the yield (40% yield), which clearly indicates that higher yields under LED irradiation (entries 1 and 2) do not arise from a thermally promoted pathway (entry 5). In fact, a high-pressure light-promoted reaction in a pressure-resistant glass tube, protected by a polycarbonate autoclave, which was cooled by a fan to keep the temperature as close as possible to room temperature (for details, see Supplementary Fig. 1). Even under 1 atm of ethylene using a balloon, the thermal ethylene insertion proceeded at a similar level, which rules out the possibility that the reaction is promoted by a potential increase of the pressure (entry 6).

Next, isolated unsymmetric **4gb** was employed to conduct the same experiment, which provided a similar trend: blue LEDs in the presence of a photocatalyst promoted the ethylene insertion in 64% yield, while white LEDs promoted the reaction in 80% yield (entries 7 and 8). These results match the yields from the 3CR (63 and 82% yield in Fig. 4). In contrast, thermal conditions under the exclusion of light in the absence of a catalyst and/or additive promoted the ethylene insertion only moderately (35% yield) and the starting diphosphine

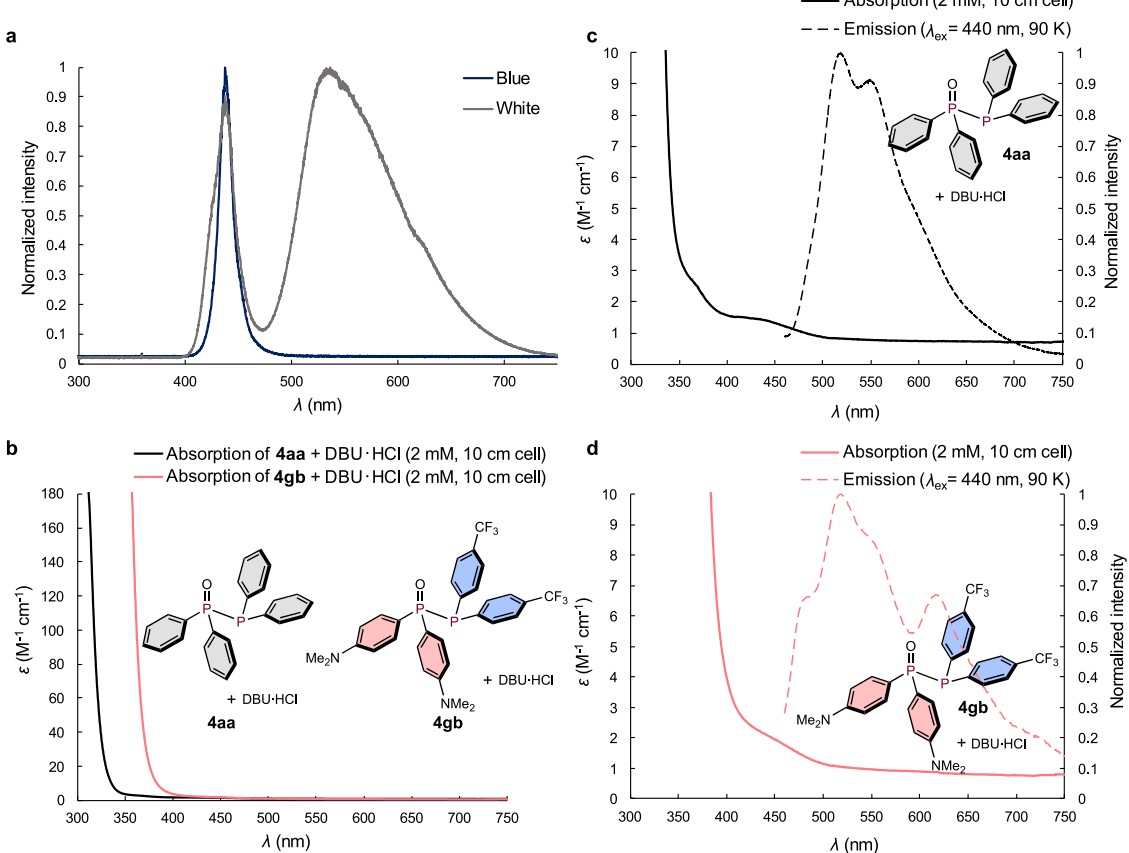

**Fig. 5 | Emission Spectra of LEDs and photophysical properties of 4. a** Emission spectra of the blue and white LEDs. **b** Absorption spectra of **4aa** and **4gb** in DCM. **c** Magnified absorption and emission spectra of **4aa** in DCM. The emission spectrum was measured at a concentration of 0.33 M. **d** Magnified absorption and emission spectra of **4gb** in DCM. The emission spectrum was measured at a concentration of 0.33 M.

**4gb** and its oxidation product **4gb′** ($p$-NMe$_2$C$_6$H$_4$)$_2$P(=O)−P(=S)($p$-CF$_3$C$_6$H$_4$)$_2$) were recovered in a combined yield of 49% yield (entry 9). For this unsymmetric substrate, photo-irradiated radical insertion is significantly more efficient than thermal insertion.

We next conducted competition experiments using isolated **4aa** and **4gb** in order to observe whether radical scrambling occurred (See Supplementary Fig. 7). Under a nitrogen atmosphere in CDCl$_3$, phosphine-scrambling was observed under irradiation from white LEDs to generate **4aa**, **4gb**, Ph$_2$P(=O)−P($p$-CF$_3$C$_6$H$_4$)$_2$ (**4ab**), and ($p$-NMe$_2$C$_6$H$_4$)$_2$P(=O)−PPh$_2$ (**4ga**) in a ratio of 34.1: 30.8: 20.5: 14.6. In contract, under complete exclusion of any light, a bit of scrambling was observed (<2%), which indicates that the generation of phosphine-centered radicals is accelerated by light, and significantly suppressed in the absence of light. Considering these experimental results, we cannot exclude at this point the possibility that a small amount of radical species kinetically formed under thermal conditions serve as radical initiators to trigger the addition of diphosphines **4** to ethylene. The addition of a radical scavenger (2,2,6,6-tetramethylpiperidine 1-oxyl (TEMPO), 1 equiv) under the exclusion of lights indeed shut down the insertion of ethylene into **4aa**, which was recovered in 92% yield. On the other hand, the evidence so far produced by experimental and computational considerations is not unequivocal that the thermal ethylene insertion proceeds via non-radical mechanisms. AFIR calculations to search a four-membered transition state of Ph$_2$P(=O)−PPh$_2$ with ethylene (ionic σ-bond metathesis process) only identified a stepwise pathway with high activation energy (ca. 50 kcal/mol).

Since the insertion of ethylene into diphosphines proceeds in the absence of any photocatalyst under irradiation from blue and white LEDs, the wavelength range of blue and white LEDs was measured experimentally to determine the peak similarities between these light sources (Fig. 5a). The white LEDs have two obvious peaks, i.e., a sharp (437 nm) and a broad peak (536 nm), which covers all wavelengths of the visible-light region. Since the blue region of the white LEDs is identical to the emission of the blue LEDs and the product yields obtained from blue and white LEDs are almost identical (entries 2 and 3 in Table 2), we assume that the crucial wavelength for promoting the 3CR is around 440 nm.

To further investigate the photophysical properties of diphosphines **4**, the absorption and emission spectra of a mixture of diphosphine **4** and DBU·HCl, which was prepared in situ from **1** and **2** in the presence of DBU (Fig. 5b-d), were measured. We first analyzed the absorption spectra of **4aa** and **4gb** by UV−Vis spectroscopy. (Fig. 5b). Even in highly concentrated solution (2 mM) using a quartz cell with a long path length (10 cm), almost no apparent absorption was observed in the visible-light region (**4aa**: av. ε$_{400-750\ nm}$ = 1.2 M$^{-1}$ cm$^{-1}$; **4gb**: av. ε$_{400-750\ nm}$ = 0.91 M$^{-1}$ cm$^{-1}$), albeit that very weak absorption was still observed in the blue region (440 nm) as evident from the magnified spectra (Fig. 5c, d), which indicates the possibility of a general S$_0$ → S$_n$ transition when using white or blue LEDs in the absence of a catalyst. Moreover, when the emission spectra of **4aa** and **4gb** were measured at 90 K under excitation at 440 nm, emission peaks were observed with an emission lifetime of 0.66 μs for the peak at 518 nm for **4aa** (dashed black line) and of 1.4 μs at 520 nm for **4gb** (dashed pink line) (Figs. 5c and 5d); based on these lifetimes, these emission peaks can be attributed to phosphorescent emission. This phosphorescence suggests the possibility that the spin-forbidden S$_0$ → T$_n$ transition is also involved based on a recently published report by Nakajima and

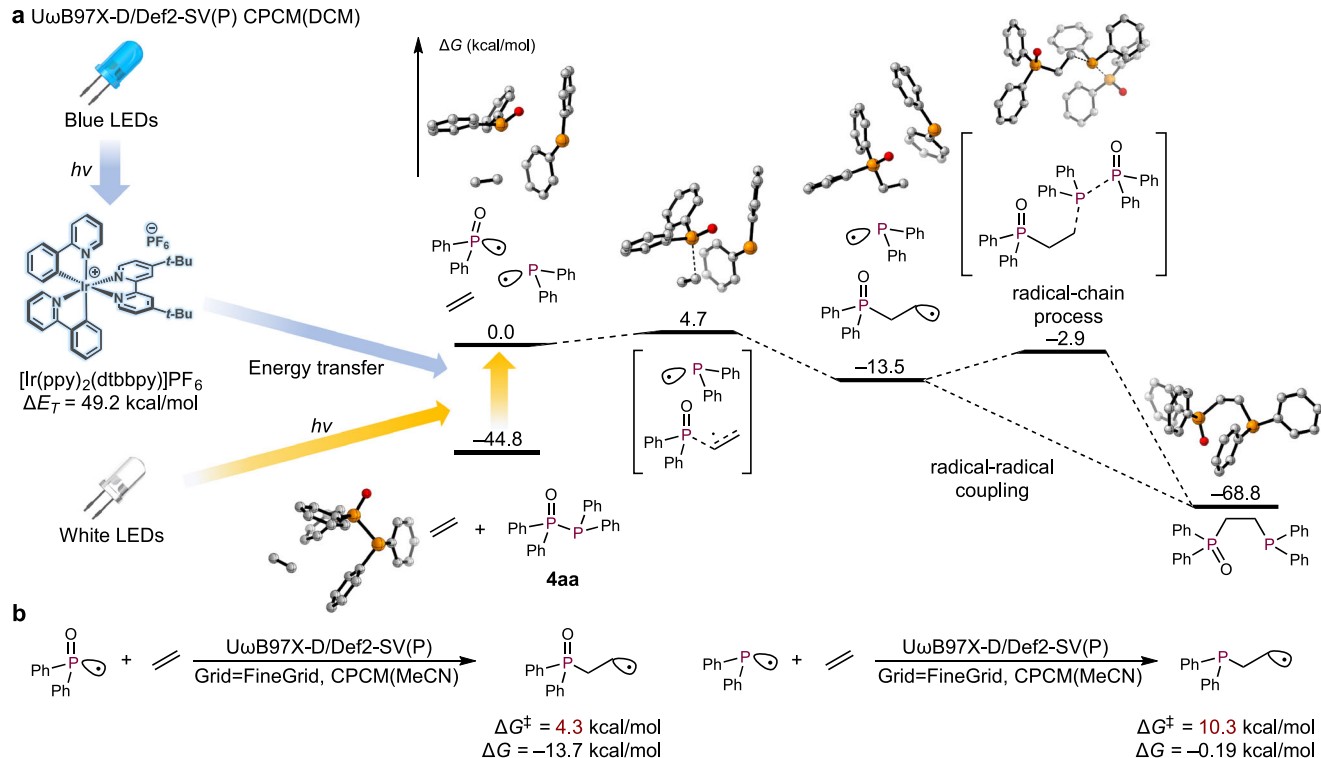

**Fig. 6 | Mechanistic studies. a** Plausible reaction mechanism. **b** Calculation results for the reaction of ethylene with phosphine-centered radicals.

Nemoto[50]. Generally, the ε values of the $S_0 \rightarrow T_n$ transition are very small, i.e., often >$10^3$ times smaller than those of the $S_0 \rightarrow S_n$ transitions[50], and the corresponding absorption peaks are thus usually difficult to detect by UV-Vis spectroscopy in the absence of heavy atoms within the molecule[50].

Based on these experimental results, we theoretically examined the light-promoted radical reaction mechanism via density functional theory (DFT) calculations using model substrate $Ph_2P(=O)-PPh_2$ (**4aa**) (Fig. 6a). Under irradiation from blue LEDs, [Ir(ppy)$_2$(dtbbpy)]PF$_6$ absorbs light at 440 nm and is excited to its triplet state ($\Delta E_T$ = 49.2 kcal/mol), which then transfers energy to **4aa** generated from $Ph_2P(O)$H (**1a**) and $Ph_2PCl$ (**2a**) in the presence of DBU, thus exciting **4aa** to the triplet state ($\Delta G$ = 44.8 kcal/mol) (Fig. 6a). Simultaneous dissociation of the P(=O)−P single bond occurs from its triplet state, and the resulting phosphoryl radical ($Ph_2P(=O)$·) reacts with ethylene to generate the terminal primary alkyl radical, which could react associatively with the remaining phosphinyl radical (radical-radical coupling). The activation energy of the addition of the phosphoryl radical to ethylene ($\Delta G^{\ddagger}$ = 4.3 kcal/mol) is smaller than that of the $Ph_2P$ radical to ethylene ($\Delta G^{\ddagger}$ = 10.3 kcal/mol), indicating that the electron-withdrawing radical initially attacks ethylene (Fig. 6b). Using white LEDs, **4aa** directly absorbs light to promote its radical dissociation via an $S_0 \rightarrow S_n$ transition and/or spin-forbidden $S_0 \rightarrow T_n$ transition (vide supra)[50]. The possibility of a radical-chain mechanism could be considered, and the energy barrier of the radical addition of the alkyl radical to another diphosphine is 10.6 kcal/mol. Even though the activation energy is much higher than that of the barrierless radical-radical coupling, the barrier can be overcome at room temperature. At this point, we cannot exclude the possibility of energetically higher radical-chain process because both the alkyl radical and the phosphinyl radical ($Ph_2P$)[51] exhibit transient character and their concentration seems to be low when considering the radical-radical coupling. Additionally, the activation energy for the addition of the alkyl radical to ethylene (13.4 kcal/mol) is higher than that to another diphosphine (10.6 kcal/mol), which reasonably explains that the radical polymerization of ethylene should

not be expected to proceed in this system. Experimentally, we did not observe the formation of any radical-polymerization products. We also considered conducting crossover experiments using two different diphosphine derivatives, i.e., **4aa** and **4gb**. If radical−radical coupling processes are dominant in the reaction, only **3aa** and **3gb** would be obtained. In contrast, if the reaction proceeds via the radical-chain mechanism, **3aa**, **3gb**, **3ab**, and **3ga** would be obtained predominantly. However, this method is unfortunately not appropriate due to the existence of a quick equilibrium between these four diphosphines (**4aa**, **4gb**, **4ab**, and **4ga**) prior to the introduction of ethylene gas (See Supplementary Fig. 7), which would render an experimental confirmation whether radical−radical coupling or a radical-chain mechanism is operative very difficult.

### Application to transition metal complexes

To examine the synthetic utility of the products, we next investigated the removal of the oxygen and sulfur atoms from product **3** in order to prepare diphosphine ligands. After oxidative exchange of S to O on the phosphine-sulfide moiety by treatment with an excess of *m*-chloroperoxybenzoic acid (*m*CPBA), the corresponding dioxides **5** were obtained in high yield for all entries. These dioxides can also be applied to the synthesis of mononuclear metal complexes and coordination polymers with lanthanide metals[2–5]. Moreover, dioxides **5** were readily reduced using the air-stable hydrosilane PhSiH$_2$-O-SiH$_2$Ph[52] in refluxing toluene to afford unsymmetric DPPEs **6** in good yield under conditions that do not require glove-box techniques (Fig. 7a). A variety of substituents with electron-donating, -withdrawing, and even bulky groups on the aromatic rings were successfully converted, and the corresponding unsymmetric DPPE dioxides **5** as well as DPPEs **6** were obtained in high yield. Furthermore, the synthesis of the symmetric DPPE derivatives with two bulky DTBM groups on each phosphorus atom succeeded via the developed two-step procedure.

We also examined a more convenient procedure for the synthesis of unsymmetric ligand **6gb**, i.e., a synthetic route that does not require oxidation using sulfur. After the 3CR of **1g** and **2b** with ethylene was

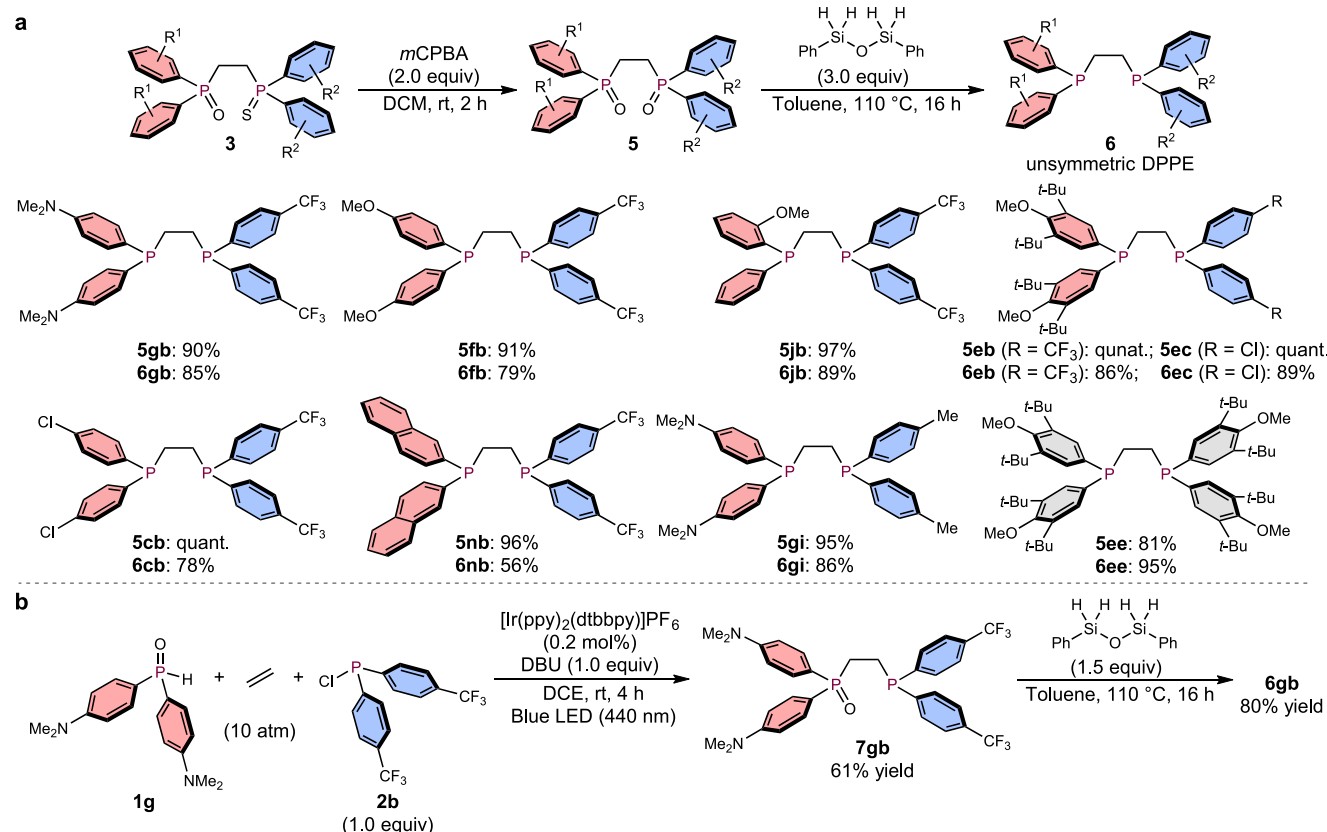

**Fig. 7 | Synthetic applications. a** Conversion of mixed P = O/P = S DPPE derivatives to DPPE dioxides, and reduction to DPPEs. **b** Alternative method for the direct reduction of monooxo DPPE derivatives.

conducted under the optimal conditions, **7gb** was isolated without oxidation using sulfur (Fig. 7b). Monooxide **7gb** was then directly reduced using the hydrosilane to afford **6gb** in 80% yield.

Having established suitable reduction conditions to synthesize DPPE derivatives, we then investigated their complexation with Ni, Pd, Pt, and Au salts (Fig. 8a). The formation of these metal complexes proceeded readily and the corresponding complexes NiCl$_2$(**6gb**), PdCl$_2$(**6gb**), PtCl$_2$(**6gb**) and Au$_2$Cl$_2$(**6gb**) were obtained in high yield. The structures of NiCl$_2$(**6gb**) and PdCl$_2$(**6gb**) were unambiguously determined using single-crystal X-ray diffraction analysis; unfortunately, single crystals of PtCl$_2$(**6gb**) and Au$_2$Cl$_2$(**6gb**) could not be obtained. Subsequently, we compared the synthesized Ni and Pd complexes with the commercially available complexes NiCl$_2$(dppe) and PdCl$_2$(dppe). Although both NiCl$_2$(**6gb**) and NiCl$_2$(dppe) exhibit similar red color (See Supplementary Fig. 13), PdCl$_2$(**6gb**) and PdCl$_2$(dppe) are yellow and white, respectively; the absorption spectrum of PdCl$_2$(**6gb**) is substantially shifted to higher wavelengths, which is responsible for the observed yellow color (Fig. 8b).

Finally, the highest occupied molecular orbitals (HOMOs) and lowest unoccupied molecular orbitals (LUMOs) of PdCl$_2$(dppe) and PdCl$_2$(**6gb**) were estimated using DFT calculations at the M06/LanL08(f) level of theory for Pd and the 6–31 G(d,p) level for all other atoms in dichloromethane (CPCM model) (Fig. 8c). In the case of PdCl$_2$(dppe), both the HOMO and LUMO are delocalized around the palladium center. Conversely, the HOMO of PdCl$_2$(**6gb**) is localized on one of the p-NMe$_2$-phenyl groups, while the LUMO is localized on the Pd center and one of the p-CF$_3$-phenyl groups. Moreover, although the energy levels of the LUMOs of PdCl$_2$(dppe) and PdCl$_2$(**6gb**) are identical, the energy level of the HOMO of PdCl$_2$(**6gb**) is higher than that of PdCl$_2$(dppe), which would explain the shift of the UV-Vis spectrum of PdCl$_2$(**6gb**) to higher wavelengths. These results indicate great

potential for such unsymmetric DPPE ligands to exhibit physical properties and reactivity patterns as auxiliary ligands in catalysis that are significantly different from those of conventional symmetric DPPE derivatives.

In summary, we have developed a facile and practical synthetic route to symmetric and unsymmetric 1,2-bis(diphenylphosphino) ethane (DPPE) derivatives from a wide range of phosphine oxides and chlorophosphines via a three-component reaction (3CR) using ethylene under irradiation from visible light (blue and white LEDs). A variety of aryl moieties substituted with groups of differing electronic/steric character can be installed on both phosphine moieties. Detailed mechanistic studies revealed that the 3CR is initiated by energy transfer from the Ir-based photocatalyst or direct light absorption of the diphosphine via a direct S$_0$ → S$_n$ transition and/or via a spin-forbidden S$_0$ → T$_n$ transition. The obtained unsymmetric DPPE derivatives can be converted into diphosphine ligands, which form complexes with Ni, Pd, Pt, and Au salts. The solid-state structures of the corresponding Ni and Pd complexes were unambiguously determined by X-ray crystallography. Interestingly, the absorption of PdCl$_2$(**6gb**) shifted to higher wavelengths, which results in a yellow color of the complex. We are now intensively studying the applications of such metal complexes to catalytic transformations. In a preliminary experiment, the catalytic asymmetric hydrogenation of methyl (Z)-2-acetamido-3-phenylacrylate using the asymmetric ligand **6jb**-Rh complex was performed under 3 atm of H$_2$. Although the yield was high (95%), the enantioselectivity was low (12% ee). Due to the ease of separation of both enantiomers by chiral HPLC at the stage of phosphine sulfides, we will continue to investigate several potential asymmetric DPPE ligands in order to improve the enantioselectivity of this reaction. We are also investigating the detailed mechanism of thermal addition to ethylene.

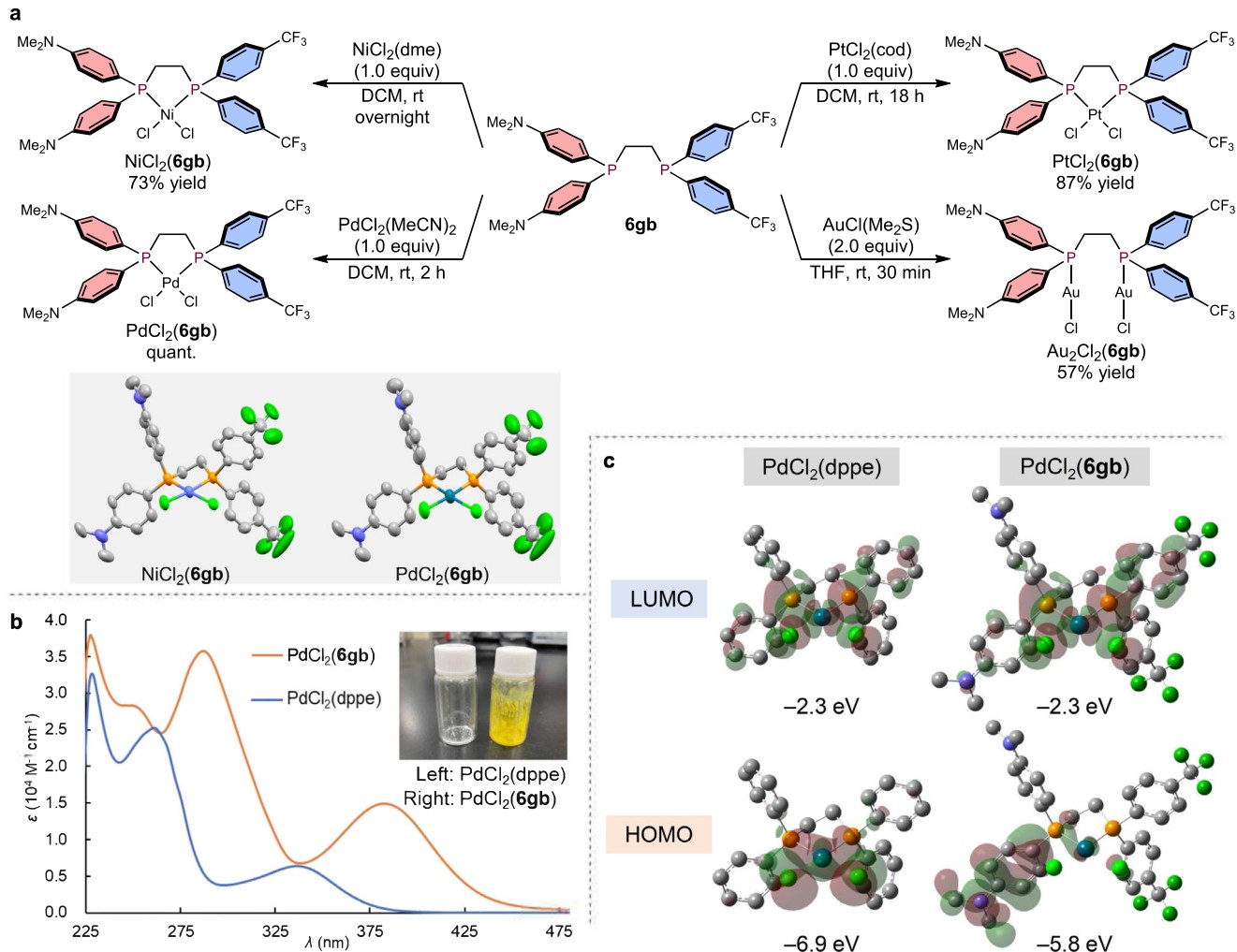

**Fig. 8 | Synthesis of transition-metal complexes and physical properties.**
**a** Synthetic application of the unsymmetric DPPE derivatives. dme: 1,2-dimethox-
yethane, cod: cycloocta-1,5-diene **b** Absorption spectra of Pd complexes
PdCl₂(**6gb**) ($5 \times 10^{-5}$ M) and PdCl₂(dppe) ($6 \times 10^{-5}$ M) in DCM. **c** HOMOs and LUMOs

of the Pd complexes PdCl₂(dppe) and PdCl₂(**6gb**). The structures were optimized at
the M06/LanL08(f) level for the Pd atoms and at the 6−31 G(d,p) level for all other
atoms. Hydrogen atoms are omitted for clarity.

## Methods

### General procedure A (for blue LED)

In an oven-dried 10 mL pressure-resistant glass tube were placed
phosphine oxide **1** (0.5 mmol, 1.0 equiv) and [Ir(ppy)₂(dtbbpy)]PF₆
(0.9 mg, 0.001 mmol, 0.2 mol%). After the addition of DCE (1.5 mL)
under nitrogen, chlorophosphine **2** (0.5 mmol, 1.0 equiv) and DBU
(74.8 μL, 0.5 mmol, 1.0 equiv) were added. Then, the tube was placed
into autoclave which consists of a polycarbonate cylinder, and ethy-
lene gas was pressurized to 10 atm. After the resulting mixture was
stirred at room temperature for 4 h under the irradiation of blue LED
(45 W PR160L-440 nm Kessil light × 2), sulfur (19.2 mg, 0.6 mmol, 1.2
equiv of S) was added into the reaction mixture. After the mixture was
stirred for 30 min, the solvent was evaporated to give the crude mix-
ture. The crude product was purified by silica-gel column chromato-
graphy to afford the product **3**.

### General procedure B (for white LED)

In an oven-dried 10 mL pressure-resistant glass tube was placed
phosphine oxide **1** (0.5 mmol, 1.0 equiv). After the addition of DCE
(1.5 mL) under nitrogen, chlorophosphine **2** (0.5 mmol, 1.0 equiv) and
DBU (74.8 μL, 0.5 mmol, 1.0 equiv) were added. Then, the tube was
placed into autoclave which consists of a polycarbonate cylinder, and
ethylene gas was pressurized to 10 atm. After the resulting mixture was

stirred at room temperature for 24 h under the irradiation of white LED
(A 160WE TUNA SUN × 2), sulfur (19.2 mg, 0.6 mmol, 1.2 equiv of S) was
added into the reaction mixture. After the mixture was stirred for
30 min, the solvent was evaporated to give the crude mixture. The
crude product was purified by silica-gel column chromatography to
afford product **3**.

## Data availability

The crystallographic data generated in this study have been deposited
in the Cambridge Crystallographic Data Centre under accession code
CCDC 2152051 (NiCl₂(**6gb**)) and 2152052 (PdCl₂(**6gb**)). Calculation
methods and cartesian coordinates of intermediates and transition
states are available in Supplementary Data 1. All of the other data
supporting the findings of this study are provided in the main text or
the Supplementary Information.

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

## Acknowledgements

The authors would like to thank Prof. Hajime Ito for fruitful suggestions. The authors would also like to express their sincerest appreciation to Ms. Yuki Konishi, Mr. Pedro Paulo Ferreira da Rosa, and Dr. Yuichi Kitagawa for helping with the analysis of the photophysical properties, as well as to Dr. Sunao Shoji for measuring the emission spectra of blue and white LEDs. Dr. Kosuke Higashida is gratefully acknowledged for performing the X-ray crystallographic analyses. This work was financially supported by JST-ERATO (JPMJER1903), JSPS-WPI, and Grants-in-Aid for Challenging Research (Exploratory) (21K18945), Scientific Research (B) (22H02069), Transformative Research Areas (A) (Digitalization-driven Transformative Organic Synthesis (Digi-TOS)) (22H05330), and Young Scientists (22K14673). T.M. thanks the Fugaku Trust for Medical Research, the Uehara Memorial Foundation, and the Naito Foundation for financial support.

## Author contributions

H.T., S.M., and T.M. conceived and designed the calculations and experiments. H.T., W.K., Y.H., and S.M. performed the calculations and analyzed the data. H.T., H.K., H.H., and T.M. performed experiments and analyzed the data. H.T. and T.M. co-wrote the manuscript with the support of H.H., W.K., Y.H., and S.M. All authors contributed to discussions.

## Competing interests

The authors declare no competing interests.
