## [Peer Review File · Nature Communications]

A theory-driven synthesis of symmetric and unsymmetric 1,2-bis(diphenylphosphino)ethane analogues via radical difunctionalization of ethyleneREVIEWER COMMENTS

Reviewer #1 (Remarks to the Author):

Maeda, Mita and co-workers present a radical mediated process to prepare unsymmetrically substituted DPPE derivatives. Their work is based on an initial AFIR study to enable reaction design. The authors use three different methods for product formation: ethene + P2Ph4; ethene + secondary phosphine + secondary chloro-phosphine; ethene + secondary phosphine oxide + secondary chloro-phosphine, opting for the latter due to ease of access to P-reagents and ease of handling. The reactions are photocatalyzed, proceeding with an Ir PC in the presence of DBU. They furnish a range of symmetrical and unsymmetrically substituted DPPE derivatives, and demonstrate that the P=O/P=S bond can be reduced in the presence of siloxane to generate the bidentate P(III) products. Mechanistic insight is offered in terms of radical trapping studies, UV-vis analysis and DFT calculations. Finally, the authors show that one of their unsymmetrical derivatives, on ligation to Pd, has a very different UV-vis profile compared to DPPE.

The article feels quite long for Nature Comm, however, if the authors have managed to remain within the word limit, then this is fine. That said, the paper is very easy to read and is ideally suited to publish in Nature Comm, with some corrections being addressed. The SI very detailed and thorough.

This reviewer appreciated the range of conditions that were studied in order to make the chemistry accessible for a wider audience e.g. blue versus white LEDs, different ethene pressures. It is interesting that the white LEDs give better yields with electron withdrawing substrates- is there a possible reason for this?

Based on the mechanism, do the authors see radical polymerisation of ethene, particularly given the radical chain process proposed and the high concentration of ethene. Is this responsible for the low yields? Do yield for some reactions increase with lower ethene loading. Based on the authors statement regarding isolating starting materials, it seems that their postulated mechanism is maybe not quite right. Could the authors calculate barriers for ethene polymerization (based on initiation from a phosphorus centred radical)? Or even a comment on how surprising that (possibly) no ethene polymerization is observed.

The aspect that lets the paper down is the poor yields, however, the authors demonstrate ease of isolation and reduction of the P(V) product.

DBU- please include full name, not just the abbreviation

Page 2, line 40- remove 'especially'

line 43- seems cursive to only include one reference, albeit a book.

Page 4, line 79- anthropomorphism, one cannot 'persuade' a molecule

Figures (e.g. Fig 1)- probably for the type-setters, but better definition between the sections a, b and c. e.g. a); b)....

Page 5, line 100- 'strong' UV light is ambiguous... maybe clearer to define wavelengths or just leave out 'strong'

Page 7, Line 159- typo "to reach the of the"

Figure 2- split into two separate figures i.e. DFT and synthesis

Page 13, line 267- phosphorus atom rather than phosphine atom

Page 16, 320- "without exclusion of light", might benefit from the inclusion of "ambient"

Page 26, line 494- typo in figure title

Reviewer #2 (Remarks to the Author):

Dppe type bidentate ligands have a wide range of applications in catalytic reactions in combination with transition metals. But synthesis of dppe ligands containing different substituents on the P atom is

still relatively rare. This article reports an efficient route for the direct preparation of symmetric and unsymmetric DPPE ligands from the reaction of ethylene, diarylphosphine oxides and chlorophosphines with a wide range of substrates and high tolerance. It is worth pointing out that Ogawa has reported the direct synthesis of dppe type ligands using diphosphane monosulfide under light (ref. 45), and both reactions follow a similar reaction mechanism. Therefore, further studies need to be conducted before publication in "Nature Communication".

- 1, New applications of unsymmetric dppe ligands needs to be elaborated;
- 2, The reaction of olefin substrates containing substituents needs to be added;
- 3, Whether existing methods can be used to synthesize dppe ligands containing alkyl substituents.

Reviewer #3 (Remarks to the Author):

In this work, the authors used visible light to synthesis symmetric and unsymmetric DPPE derivatives from a wide range of phosphine oxides and chlorophosphines via a three-component reaction using ethylene. The authors have done the experiments including the aryl 27 moieties substituted with groups of differing electronic/steric character installed on both phosphine moieties and the DFT calculations on mechanism studies, and the results are good. But it is noted that the work is not sufficient in novelty in publishing it in Nat Commun because the reactions have been studied previously by other methods, I recommend its publication in journals like Org Lett, Chem Commun or Chem-Eur J.

Point-by-point Response to the Reviewers' Comments

Responses to referee #1:

- 1) The article feels quite long for Nature Comm, however, if the authors have managed to remain within the word limit, then this is fine. That said, the paper is very easy to read and is ideally suited to publish in Nature Comm, with some corrections being addressed. The SI very detailed and thorough.

Thank you very much for the valuable feedback. We have moved Figure 5 to the SI (Figure S4) in order to reduce the volume of the paper (in total 10 figures). The following figure numbers have been adjusted to reflect this change. The main text (excluding figure legends and Methods) should not be longer than 6,000 words according to the author guidelines ([chrome-extension://efaidnbmnnnibpcajpcgclefindmkaj/https://www.nature.com/documents/ncomms-formatting-instructions.pdf](https://www.nature.com/documents/ncomms-formatting-instructions.pdf)). The current word count of this manuscript is ca. 5,300, which is within the stipulated word limit.

- 2) This reviewer appreciated the range of conditions that were studied in order to make the chemistry accessible for a wider audience e.g. blue versus white LEDs, different ethene pressures. It is interesting that the white LEDs give better yields with electron withdrawing substrates- is there a possible reason for this?

The photocatalyst does not only exhibit energy-transfer properties, but also photoredox activity to oxidize and/or reduce substrates and thus promote side reactions. Electron-withdrawing substrates seem to be more sensitive to photoredox-catalyzed reduction conditions that trigger several side reactions, thus leading to the decreased yield.

- 3) Based on the mechanism, do the authors see radical polymerisation of ethene, particularly given the radical chain process proposed and the high concentration of ethene. Is this responsible for the low yields? Do yield for some reactions increase with lower ethene loading. Based on the authors statement regarding isolating starting materials, it seems that their postulated mechanism is maybe not quite right. Could the authors calculate barriers for ethene polymerization (based on initiation from a phosphorus centred radical)? Or even a comment on how surprising that (possibly) no ethene polymerization is observed.

Ethylene-polymerization products were not obtained in this reaction system. The activation energy of the addition of the primary alkyl radical, which is generated from the phosphorus radical, to ethylene (13.4 kcal/mol) is higher than that to another diphosphine (10.6 kcal/mol), which reasonably explains that the radical polymerization of ethylene does not proceed in this system.

We have added the following sentence on page 22, line 433: "The activation energy for the addition of the alkyl radical to ethylene (13.4 kcal/mol) is higher than that to another diphosphine (10.6 kcal/mol), which reasonably explains that the radical polymerization of ethylene should not be

expected to proceed in this system. Experimentally, we did not observe the formation of any radical-polymerization products.”

4) DBU- please include full name, not just the abbreviation

The formal chemical name (1,8-diazabicyclo[5.4.0]undec-7-ene) has been added.

Page 2, line 40- remove ‘especially’

“Especially” has been removed.

line 43- seems cursive to only include one reference, albeit a book.

The referenced book has been replaced with an appropriate paper: van Leeuwen, P. W. N. M., Kamer, P. C. J., Reek, J. N. H. & Dierkes, P. Ligand bite angle effects in metal-catalyzed C–C bond formation. *Chem. Rev.* **100**, 2741–2770 (2000).

Page 4, line 79- anthropomorphism, one cannot ‘persuade’ a molecule

“Persuade” has been removed and the sentence been modified to: “If the two carbon atoms of ethylene could react with two phosphine-centered radicals...”.

Figures (e.g. Fig 1)- probably for the type-setters, but better definition between the sections a, b and c. e.g. a); b)....

The style of the sections (a, b, c...) in all figures was changed to match the figure legends.

Page 5, line 100- ‘strong’ UV light is ambiguous... maybe clearer to define wavelengths or just leave out ‘strong’

“Strong” has been removed.

Page 7, Line 159- typo “to reach the of the”

“The” has been removed between ‘reach’ and ‘of’.

Figure 2- split into two separate figures i.e. DFT and synthesis

We have arranged Figure 2 according to the request.

Page 13, line 267- phosphorus atom rather than phosphine atom

“Phosphine” has been changed to “phosphorus”.

Page 16, 320- “without exclusion of light”, might benefit from the inclusion of “ambient”

“Without exclusion of light” has been changed to “with complete exclusion of ambient light”

Page 26, line 494- typo in figure title

“Synythesis” has been changed to “Synthesis”.

Reviewer #2 (Remarks to the Author):

1) New applications of unsymmetric dppe ligands needs to be elaborated;

In a preliminary experiment, the catalytic asymmetric hydrogenation of methyl (Z)-2-acetamido-3-phenylacrylate using the asymmetric ligand **6jb**-Rh complex was performed under 3 atm of H₂. Although the yield was high (95%), the enantioselectivity was low (12% ee). Due to the ease of separation of both enantiomers by chiral HPLC at the stage of phosphine sulfides, we will continue to investigate several potential asymmetric DPPE ligands in order to improve the enantioselectivity

of this reaction. We have added this information to the conclusion section on page 27, line 521. Moreover, we have employed diphosphine oxide **5gb** as a unsymmetric ligand for the synthesis of a lanthanoid-based coordination polymer, which shows temperature-depended luminescence properties around room temperature. These results will be disclosed elsewhere in due course.

The reaction of olefin substrates containing substituents needs to be added;

We have already tested styrene and 1-hexene (3 equiv) instead of ethylene, but these reaction systems are complicated and target compounds were not observed in either case, which indicates that this 3CR is highly selective toward ethylene. This has already been mentioned on page 14, line 293. As for the further extension of our synthetic methodology, we are currently applying our 3CR to highly strained small cyclic compounds such as [1.1.1]propellane, which produces a diphosphine-containing bicyclo[1.1.1]pentane motif in high yield. These results will be published elsewhere in due course.

- 2) Whether existing methods can be used to synthesize dppe ligands containing alkyl substituents. We also examined the 3CR of dialkylphosphine oxides (*n*-butyl, *tert*-butyl, and cyclohexyl), chlorodiarylphosphines, and ethylene; however, these reaction systems were complicated and the target compounds were not obtained. We have added this information on page 14, line 296.

Reviewer #3 (Remarks to the Author):

- 1) In this work, the authors used visible light to synthesis symmetric and unsymmetric DPPE derivatives from a wide range of phosphine oxides and chlorophosphines via a three-component reaction using ethylene. The authors have done the experiments including the aryl 27 moieties substituted with groups of differing electronic/steric character installed on both phosphine moieties and the DFT calculations on mechanism studies, and the results are good. But it is noted that the work is not sufficient in novelty in publishing it in Nat Common because the reactions have been studied previously by other methods, I recommend its publication in journals like Org Lett, Chem Commun or Chem-Eur J.

The related methods reported thus far do not employ ethylene. In particular, Hirano and Miura's protocol with a photocatalyst in combination with NBS (ref. 44) shows that "attempts to apply aliphatic alkenes such as oct-1-ene remained unsuccessful (trace, data not shown)". Therefore, our method is characterized by heigh levels of utility for the synthesis of DPPE derivatives, given that simple aliphatic alkenes such as ethylene can be used as a starting material. Furthermore, all precedents reported employ unstable diphosphines as starting materials; in contrast, our method employs two stable compounds, i.e., phosphine oxides and chlorophosphines, which is of significant practical value compared to other methods.

REVIEWERS' COMMENTS

Reviewer #1 (Remarks to the Author):

The authors have addressed all comments and requested revisions. As a result, this manuscript is suitable for publication in Nature Communications in its current form.

Point-by-point Response to the Reviewers' Comments

Responses to referee #1:

The authors have addressed all comments and requested revisions. As a result, this manuscript is suitable for publication in Nature Communications in its current form.

We appreciate the helpful comments.